

# Effects of low-frequency vibration training on walking ability and body composition among older adults: a randomized controlled trial

Xiaohuan Tan[1], Guiping Jiang[2], Lei Zhang[1], Dandan Wang[3] and Xueping Wu[1]

[1] Shanghai University of Sport, Shanghai, China
[2] Harbin University, Harbin, China
[3] University of Shanghai for Science and Technology, Shanghai, China

Corresponding author
Xueping Wu, wuxueping@sus.edu.cn

## ABSTRACT

**Background.** As life expectancy rises, age-related decline in mobility and physical function poses challenges for older adults. While traditional exercise can help, limitations and injury risks persist. This study explores low-frequency vibration training as a potential alternative to improve walking ability and body composition in older adults.

**Methods.** A lottery was used to randomly assign 50 participants (mean age 80.08 years) to either a vibration group ($n = 25$, 10 males, 15 females) or a control group ($n = 25$, 11 males, 14 females). While the control group continued their regular daily schedule, the vibration group completed 8 weeks of low-frequency vibration training (frequency: 4–13 Hz; amplitude: two mm), three sessions per week, with each session lasting 20–30 minutes. The walk ability was assessed using the 30-second Chair Stand Test (30-s CST), Timed Up and Go (TUG), and six-meter (six m) walk speed, while body composition was measured via body mass index (BMI), body fat percentage, and waist circumference (WC), hip circumference (HC), and waist-to-hip ratio (WHR).

**Results.** Low-frequency vibration training significantly increased walking speed in the six m walk speed ($F_{(1,36)} = 4.50$, $p = 0.04$, $\eta_p^2 = 0.11$) and TUG ($z = -2.72$, $p = 0.007$), compared with the control group. Observed improvements on the 30-s CST were not statistically significant ($F_{(1,36)} = 0.05$, $p = 0.81$, $\eta_p^2 = 0.002$). In the WC, the effect of time ($F_{(1,36)} = 7.19$, $p = 0.01$, $\eta_p^2 = 0.16$) was significant. The main effect of the group for HC ($F_{(1,36)} = 0.06$, $p = 0.80$, $\eta_p^2 = 0.002$) and WHR ($F_{(1,36)} = 2.00$, $p = 0.16$, $\eta_p^2 = 0.05$) were not significant, but the interaction effects for HC ($F_{(1,36)} = 6.37$, $p = 0.01$, $\eta_p^2 = 0.15$) and WHR ($F_{(1,36)} = 9.08$, $p = 0.005$, $\eta_p^2 = 0.20$) were significant. However, the intervention showed no statistically significant effects on BMI and body fat percentage.

**Conclusion.** Low-frequency vibration training significantly enhanced walking speed and WHR in older adults. This low-intensity intervention is especially beneficial for those with exercise limitations or a high risk of injury. Although its effects on BMI and body fat percentage were limited, the study offers valuable insights for developing personalized vibration training programs.

# INTRODUCTION

Age-related disorders are rapidly turning into a global issue that cannot be disregarded as the average life expectancy of humans rises due to advancements in social infrastructure and medical technology (*López-Otín et al., 2013*). The World Health Organization Global Report on Ageing and Health states that an individual's inherent capacity affects how well they can physically perform (*World Health Organization, 2015*). Intrinsic capacity is defined as "the combination of all bodily functions and brain power that an individual can use at any given time" ,whereas intrinsic capacity decline is primarily characterized by issues such as decreased mobility, malnutrition, visual and auditory impairment, cognitive disorders, and depressive symptoms (*World Health Organization, 2017*). Mobility is a crucial aspect of physical function in older adults, and independent and autonomous walking is a basic requirement for people to display functional performance (*Kim et al., 2013*).

According to recent studies, loss of muscular strength, decreased functional mobility, and balance issues are all significant factors that affect mobility (*Tian et al., 2016*). A decline in physical function, particularly at advanced ages, is manifested as a significant decline in lower limb motor function (*Mitra & Sambamoorthi, 2014*). Deterioration in walking ability among older adults becomes increasingly apparent when they face more interconnected physical and mental health problems (*Verghese et al., 2009*). As people age, it is crucial to ensure that they can walk independently (*Mamikonian-Zarpas & Laganá, 2015*). Increased mobility benefits older people's ability to exercise, boosts immunity, lowers medical costs, and lowers death rates, which in turn enhance their quality of life and ease major social issues and financial obligations (*World Health Organization, 2015*). However, research among older adults involving aspects such as walking have been overlooked in recent studies despite an immediate need to address how to prevent functional decline in older adults to enhance their walking ability.

The necessity for the lower limbs to support the body's weight—another important aspect in determining an older person's capacity to walk—presents one of the largest barriers for older adults. Only 2.5%–22% of older adults meet the current World Health Organization recommendations for physical activity (150 min of moderate intensity physical per week) (*Sagelv et al., 2019*). The results of a previous study comparing the ratio of lower limb strength to body mass index (BMI) found that this ratio gradually decreases with age, and is followed by a decreased ability to walk (*Tanaka et al., 2020*; *Koushyar et al., 2020*). A lack of sufficient exercise among older adults may lead to too much fat and too little muscle, and this change in body composition may affect the capacity to accomplish daily duties.

Resistance, endurance, and multicomponent exercise training have been shown to slow or even make up for the loss of age-related physical performance and physiological decline as well as to boost the body's metabolism and increase fat consumption (*Chodzko-Zajko et al., 2009*). However, older persons with physical limitations brought on by a variety of physical issues (such as heart problems or loss of balance) may not only lack the desire to be active and exercise owing to a monotonous exercise routine and requirement for

supervision throughout training, but also may lack the ability to do so. Additionally, physical limitations may make injuries more likely (*Sousa et al., 2014*).

Vibration training is be an efficient and simple therapeutic approach (*Tsuji et al., 2014*) that has gained popularity for use in sports fields because it improves the function of various body organs, and it has been shown to receive a high level of adherence from older individuals (*Wadsworth & Lark, 2020*; *Tan et al., 2023*). It is mechanical stimulation from a vibration platform that activates $\alpha$ motor neurons *via* a monosynaptic pathway to initiate muscle contraction, while simultaneously stimulating abdominal proprioceptors to enhance contraction force—factors that are critical for walking performance (*Rittweger, 2010*). However, relatively few studies have assessed the effect of vibration training on the ability of older adults to walk, a basic requirement for many physical activities. Vibration training has been shown to improve muscle performance and balance, and fall prevention in middle-aged and older adults (*Goudarzian et al., 2017*; *Wei et al., 2017*; *Tan et al., 2023*). However, the safety and efficacy of high-frequency vibration training in this population remain controversial (*Machado et al., 2010*; *Saucedo et al., 2021*; *Greco et al., 2024*). In contrast, a study found that low-frequency vibration training provides accessible and sufficient stimulation for frail older adults to achieve improvements in physical function (*Wadsworth & Lark, 2020*). Nevertheless, it is currently unclear whether low-frequency vibration training can achieve comparable effects to high-frequency protocols (*Sievänen et al., 2014*; *Tseng et al., 2021*). Thus, this study investigated the effects of low-frequency vibration training on the walking ability and body composition of older adults.

## MATERIALS AND METHODS

### Study design
Participants were randomly assigned to either the experimental or control groups using a lottery method, without stratifying by sex. Group allocation cards labeled "experimental" or "control" were placed in a concealed container, and participants drew one card to determine their group assignment. The process was conducted under the supervision of a single supervisor, who ensured the proper implementation of the randomization procedure. Moreover, to reduce bias, outcome assessors were blinded to the participants' group assignments. All participants voluntarily provided written informed consent, and personal information was kept confidential to ensure privacy and security.

### Participants
Conducted using G*Power (version 3.1), the a priori power analysis determined that a minimum sample size of 34 participants is required to achieve 80% power for a medium effect size ($f = 0.25$) in a $2 \times 2$ repeated measures ANOVA (*Cohen, 1988*). This calculation aligns with a significance level of $\alpha = 0.05$, following *Cohen*'s (*1988*) guidelines. All participants were residents of Shanghai, China, at least 75 years of age, and could execute the tasks and walk independently. Exclusion criteria were (1) stroke, severe cardiac disease, or stent placement that was not appropriate for the level of required exercise; (2) spinal cord injury, endogenous osteosynthesis, knee or hip replacement, pacemaker, cardiovascular illness, and epilepsy; and (3) osteoarthritic conditions that limited ability to exercise,

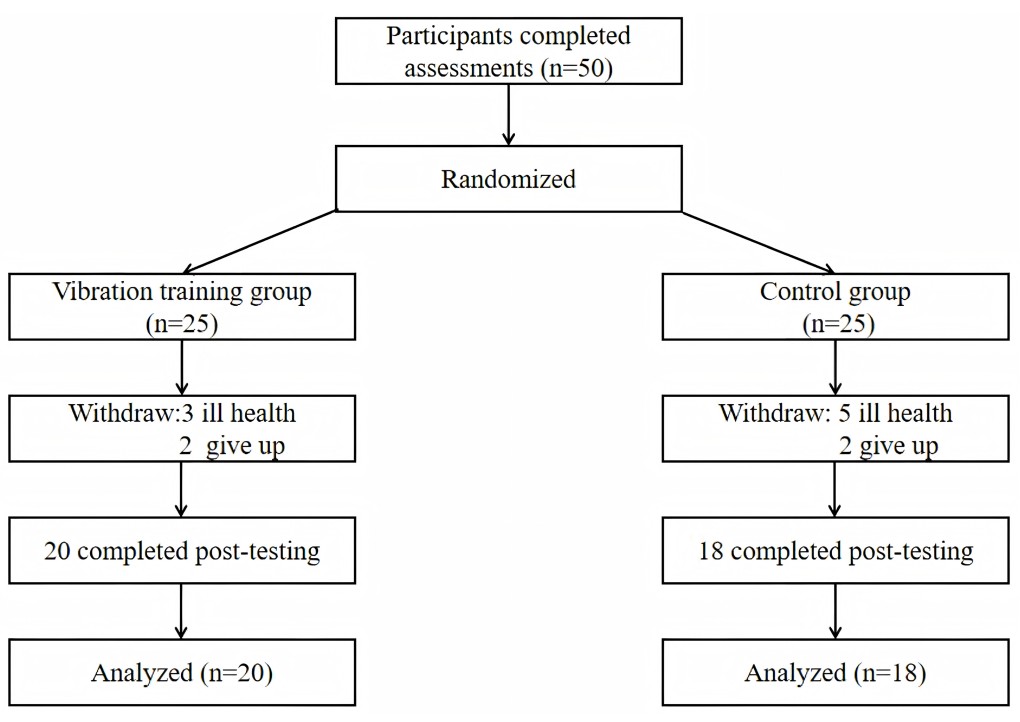

**Figure 1  CONSORT flow diagram.**

osteoporosis, and physical or cognitive disorders that would hamper training and testing procedures. Participants were randomly assigned to vibration group ($n = 25$) and control group ($n = 25$). A drop out in vibration group occurred due to health problems ($n = 3$) and loss of interest ($n = 2$) and in the control group due to personal reasons ($n = 2$), health problem ($n = 5$) (Fig. 1). The study was approved by the Ethics Committee of Shanghai University of Sport (102772022RT123).

## Intervention

Participants in the vibration group stood on a vibration platform (Body Green, Taiwan) three times a week for 8 weeks while maintaining a knee flexion angle of 40–60 degrees (Fig. 2). The platform had handrails for safety. Based on previous studies (*Wadsworth & Lark, 2020*), we designed the vibration intervention protocol for this study. In our protocol, the vibration frequency for each session was initially set at 5 Hz and then gradually increased over the 8-week period (Table 1). This design aims to provide participants with progressively enhanced stimulation, thereby ensuring safety while improving the effectiveness of the training (*Wadsworth & Lark, 2020*). The vibration amplitude was maintained at two mm. The vibration frequency used for each session began at 5 Hz but then increased across the 8 weeks as shown in Table 1. The vibration lasted for 1 min each bout, starting with five bouts per session for weeks 1–4 and then increasing to 10 bouts for weeks 5–8. Participants rested for 1 min on the platform between vibration bouts. After vibration training, participants performed 5 min of stretching and relaxation exercises,

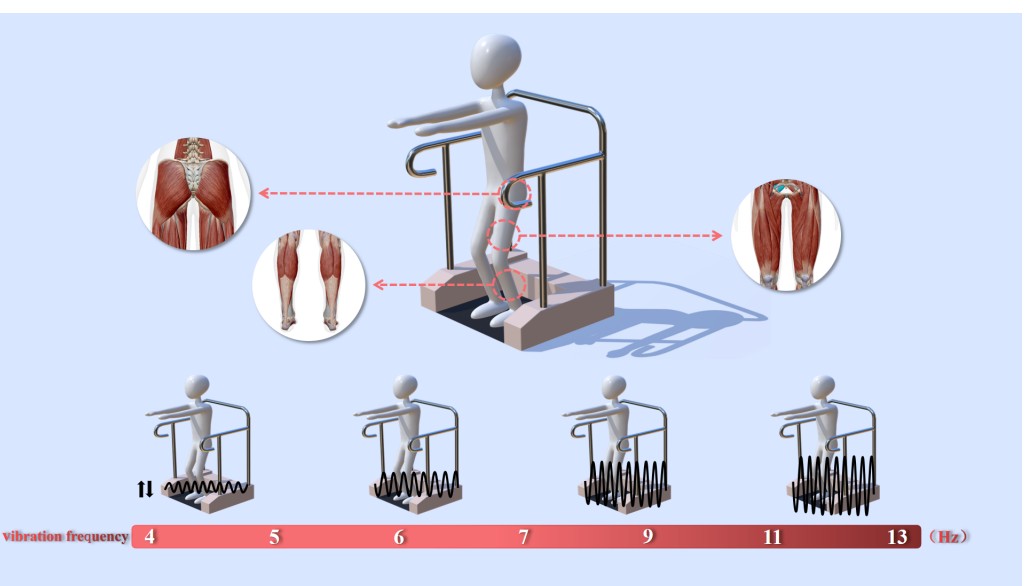

**Figure 2 Schematics of the vibration platform.** NOTE: The frequency was gradually increased to increase the vibration stimulation.

**Table 1 Vibration training program variables.**

|                          | Weeks 1–2 | Weeks 3–4 | Weeks 5–6 | Weeks 7–8 |
|--------------------------|-----------|-----------|-----------|-----------|
| Vibration frequency (Hz) | 5–7       | 5–9       | 5–11      | 5–13      |
| No. of bouts (1 min each)| 5         | 5         | 10        | 10        |
| Time between bout (s)    | 60        | 60        | 30        | 30        |
| Total time (min)         | 20        | 20        | 30        | 30        |

including two sets of static stretching for the lower limbs (hamstrings, quadriceps, and calves), each held for 15 s per side, followed by two minutes of diaphragmatic breathing exercises to promote relaxation. The control group maintained their normal daily routine, with activities of daily living monitored by an investigator on a weekly basis.

## Walking ability

Walking ability is defined as the capacity to generate displacement by utilizing lower limb muscle strength to support body weight against gravity and provide the necessary forward and upward propulsion during ambulation. This ability facilitates critical transitional movements while maintaining dynamic balance by keeping the center of gravity within the base of support (*Suwannarat et al., 2021*). Walking ability was evaluated using the 30-second Chair Stand Test (30-s CST), Timed Up and Go (TUG) test and the 6 m walk speed.

The 30-s CST is the most widely used test method for assessing lower limb muscular strength and its role in transitional movements during ambulation, such as sit-to-stand and stand-to-walk transitions among older individuals in a community (intraclass correlation coefficient (ICC), 0.84 for men, and 0.92 for women) (*Barrios-Fernández et al., 2020*).
Participants sat in a chair with the back straight, feet shoulder-width apart, knees bent at 90 degrees, and arms crossed in front of the chest. The number of times that a participant stood from a seated position and sat down again within 30 s was recorded. The average of two tests was used to calculate the final exam score (*Jones, Rikli & Beam, 1999*).

The TUG test is a dynamic balance and mobility test with a high rate of retest reliability (ICC = 0.99) (*Suwannarat et al., 2021*). TUG measures a person's ability to maintain dynamic balance while moving about and changing positions. Participants sat in a chair until a tester said "go", at which time the participant stood and walked three m at their usual walking speed before returning to sit in the chair (*Gadelha et al., 2018*). The tester noted any assistance needed and recorded the time in seconds it took to complete the test. The average of two measurements used to determine the outcome.

Walking speed, which has a high retest reliability (ICC of 0.92) (*Lyons et al., 2015*), provides the strongest diagnostic metric for evaluating walking ability in older people (*Tai et al., 2021*). The first and last two m of a 10-m open space are not included in the test to avoid acceleration and deceleration effects. Participants walked 10 m from a marked starting line to a marked finish line. When the participant's front toe crossed the two-meter line, the tester started a stopwatch and stopped it when the participant's toe reached the eight-m line (*Zhao, Wu & Wei, 2020*). Participants were asked to walk at their maximum pace. The final score was determined by averaging the results of two tests.

## Body composition

Body composition was evaluated using the height and weight, body mass index (BMI) and body fat percentage, and circumference. Height and weight were obtained using an SH-200 automatic ultrasonic height and weight measuring device (also known as a medical height and weight scale), an electronic and an electronic scale from AVIC Precision Pressure Sensors to measure weight. BMI and body fat percentage were determined through bioelectrical impedance analysis using a body composition analyzer (Tanita Corporation, MC-980MA, Tokyo, Japen). The instrument has a total of eight contact electrodes to assess a variety of bodily signs quickly enough to perform the test. A soft leather tape measure was used to measure the circumference of the waist and hips to one decimal point (0.1 cm). The circumference measurement method estimates body fat content through the measurement of the circumference of the body parts where subcutaneous fat accumulates more, typically the waist circumference (WC) and hip circumference (HC). WC divided by HC is the waist-to-hip ratio (WHR).

## Statistical analyses

Independent samples were assessed, and the findings were analyzed using SPSS version 26.0 software (IBM Corp., Armonk, NY, USA). The pre-intervention baseline data were analyzed using a $t$-test. The temporal within-group effects and differences between the vibration and control groups were examined using repeated-measures analyses of variance (ANOVAs). If there was a statistically significant interaction between group and time, the data were further analyzed using simple effects test. If the data fail to meet the assumption of normal distribution, non-parametric tests should be utilized for statistical analysis. Effect

size classifications: trivial ($\eta_p^2 < 0.01$, minimal impact), small ($0.01 \leq \eta_p^2 < 0.06$, weak), medium ($0.06 \leq \eta_p^2 < 0.14$, moderate), and large ($\eta_p^2 > 0.14$, substantial) (*Cohen, 1988*). A significance level of $p < 0.05$ was established.

# RESULTS

## Participant characteristics
Demographic characteristics for the two participant groups are presented in Table 2 and Table 3. Comparisons of age, walking ability, and body composition between the cohorts revealed no statistically significant differences, indicating that the participant groups were equivalent for the purposes of this study.

## Effect of vibration training on walking ability of older adults
With group as the between-subjects factor (VG and CG) and time as the within-subjects factor, a two-way repeated measures ANOVA was conducted to investigate the trend of variation. In the 30-s CST, the results showed that the main effect of the group ($F_{(1,36)} = 0.05$, $p = 0.81$, $\eta_p^2 = 0.002$) was not significant (Fig. 3). In walking speed, the main effect of the group ($F_{(1,36)} = 4.50$, $p = 0.04$, $\eta_p^2 = 0.11$) was significant. In TUG test, using the non-parametric Mann–Whitney test, after the intervention, there was a significant difference between the experimental group and the control group ($z = -2.72$, $p = 0.007$) (Fig. 4).

## Effects of vibration training on body composition
In the WC, the results revealed that the main effect of the group ($F_{(1,36)} = 0.80$, $p = 0.37$, $\eta_p^2 = 0.02$) and the interaction effect ($F_{(1,36)} = 1.96$, $p = 0.16$, $\eta_p^2 = 0.05$) were not significant. Moreover, the effect of time ($F_{(1,36)} = 7.19$, $p = 0.01$, $\eta_p^2 = 0.16$) was significant. The main effect of the group for HC ($F_{(1,36)} = 0.06$, $p = 0.80$, $\eta_p^2 = 0.002$) and WHR ($F_{(1,36)} = 2.00$, $p = 0.16$, $\eta_p^2 = 0.05$) were not significant, but the interaction effects for HC ($F_{(1,36)} = 6.37$, $p = 0.01$, $\eta_p^2 = 0.15$) and WHR ($F_{(1,36)} = 9.08$, $p = 0.005$, $\eta_p^2 = 0.20$) were significant. The main effect of the group for the WHR ($F_{(1,36)} = 2.00$, $p = 0.16$, $\eta_p^2 = 0.05$) (Fig. 5). The main effect of the group for the BMI ($F_{(1,36)} = 0.84$, $p = 0.36$, $\eta_p^2 = 0.02$) and body fat percentage ($F_{(1,36)} = 0.48$, $p = 0.48$, $\eta_p^2 = 0.01$) were not significant (Fig. 6).

# DISCUSSION

This randomized controlled trial examined the physiological effects of 8 weeks of low-frequency vibration training on dynamic balance, walking ability, and body composition of older adults to assess whether low-frequency vibration training was safe and effective in this population. Our key findings indicated that 8 weeks of low-frequency vibration may help to preserve lower limb muscle strength and significantly enhanced both dynamic balance and maximum walking speed as well as improved body composition. There were no adverse effects of the training. Thus, long-term low-frequency vibration training may be useful as a safe and effective tool for improving the health of older adults.

**Table 2  Baseline characteristics of participants (M ± SD).**

| Parameters | VG (n = 20) | CG (n = 18) | t | p |
|---|---|---|---|---|
| Age (y) | 79.00 ± 3.85 | 81.28 ± 4.18 | −1.73 | 0.09 |
| BMI (kg/m²) | 23.76 ± 3.60 | 24.48 ± 3.04 | −0.66 | 0.51 |
| Body fat (%) | 29.47 ± 8.93 | 30.94 ± 9.87 | −0.48 | 0.63 |
| 30s CST (counts) | 15.15 ± 4.71 | 16.27 ± 6.93 | 0.59 | 0.55 |
| 6 m walk speed (m/s) | 1.19 ± 0.24 | 1.29 ± 0.34 | −0.07 | 0.30 |
| WC (cm) | 89.14 ± 9.31 | 90.07 ± 8.80 | −0.52 | 0.60 |
| HC (cm) | 97.66 ± 6.73 | 100.56 ± 6.52 | −1.34 | 0.81 |
| WHR | 0.91 ± 0.05 | 0.90 ± 0.05 | 0.58 | 0.56 |

**Notes.**
All values represent mean ± SD; BMI, body mass index; CG, control group; 30-s CST, 30-second Chair Stand Test; HC, hip circumference; VG, vibration group; WC, waist circumference; WHR, waist-to-hip ratio.

**Table 3  Baseline characteristics of participants M (P25, P75).**

| Parameter | VG (n = 20) | CG (n = 20) | z | p |
|---|---|---|---|---|
| TUG (s) | 9.23 (8.50, 13.00) | 8.60 (8.19, 10.46) | −2.19 | 0.214 |

**Notes.**
The TUG (Timed Up and Go) data do not conform to a normal distribution; therefore, non-parametric tests are employed for statistical analysis. M, the median; P25, the 25th percentile; P75, the 75th percentile; z, Mann–Whitney U test; VG, vibration group; CG, control group

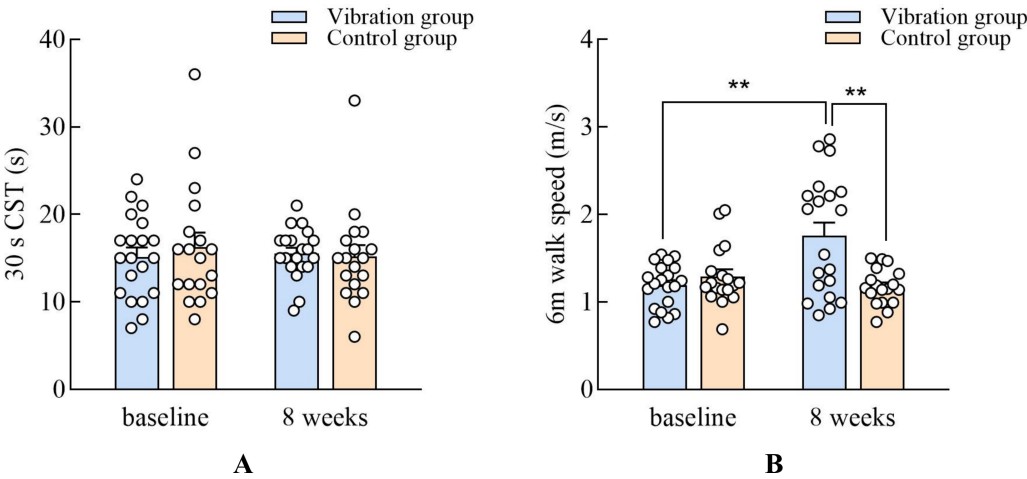

**Figure 3  Comparisons of the change after 8 weeks training in the the 30-s CST and six m walk speed.**
NOTE: *, $P < 0.05$; **, $P < 0.01$.

## Effects of vibration training on walking ability of older adults

The number of counts in the 30-s CST slightly increased in the vibration training group, while a decrease was observed in the control group. However, these changes did not reach statistical significance, indicating that the vibration training did not significantly improve lower limb muscle strength or endurance as measured by the 30-s CST. A study by *Zhu et al. (2019)* assessed lower limb muscle strength among older men (88.5 ± 3.7 years old) who

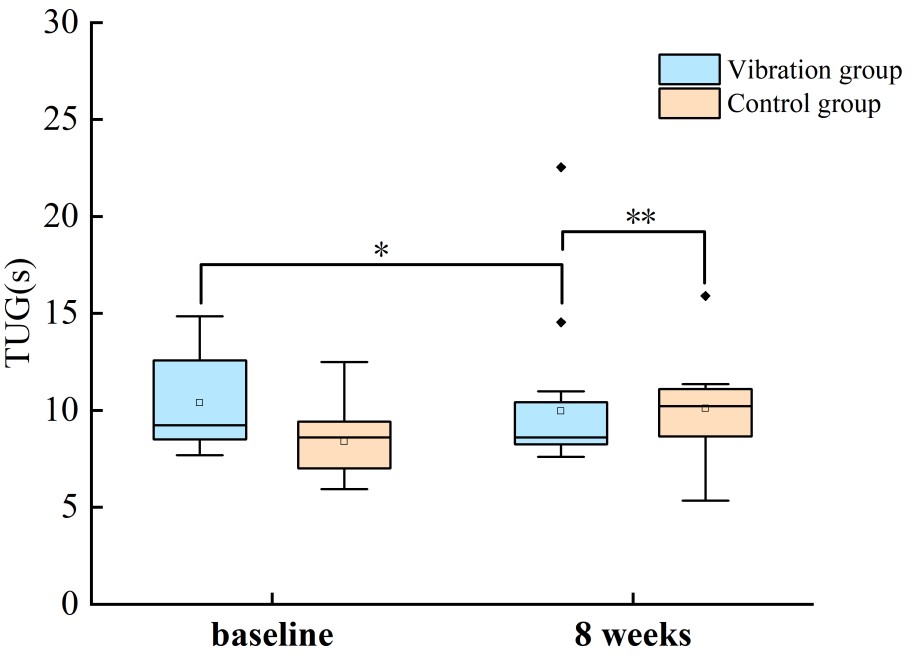

**Figure 4 Comparisons of the change after 8 weeks training in the TUG.** NOTES: The box plot illustrates the median (horizontal line within the box), mean (small square within the box), interquartile range (IQR, represented by the box), and the maximum and minimum values (whiskers). Outliers are represented by individual points outside the whiskers. TUG, Timed Up and Go; *, $P < 0.05$; **, $P < 0.01$.

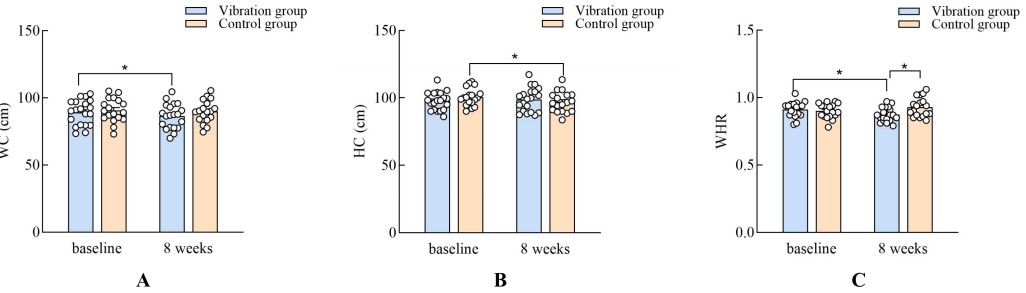

**Figure 5 Comparisons of the change after 8 weeks training in the WC, HC and WHR.** NOTES: WC, waist circumference; HC, hip circumference; WHR, waist-to-hip ratio; *, $P < 0.05$.

held a vibrating rope with both hands and flexed their knees while slightly squatting after 5 weeks of vibration training (12–16 Hz, 3–5 mm, 5 times/week) (*Zhu et al., 2019*). Their results showed that whole-body vibration training group significantly improved lower limb muscle strength, with lower limb strength increased by 15.6%–18.2% following the intervention. The authors indicated that there may be gender differences in the effects of vibration training. A study by *Faes et al. (2018)* found that muscle activity did not increase with vibration training. However, that study assessed only the acute effects, which may not have provided sufficient time for the stimulation to make a difference. In addition, the participants in the study by *Faes et al. (2018)* were young adults (20 years or older), and the

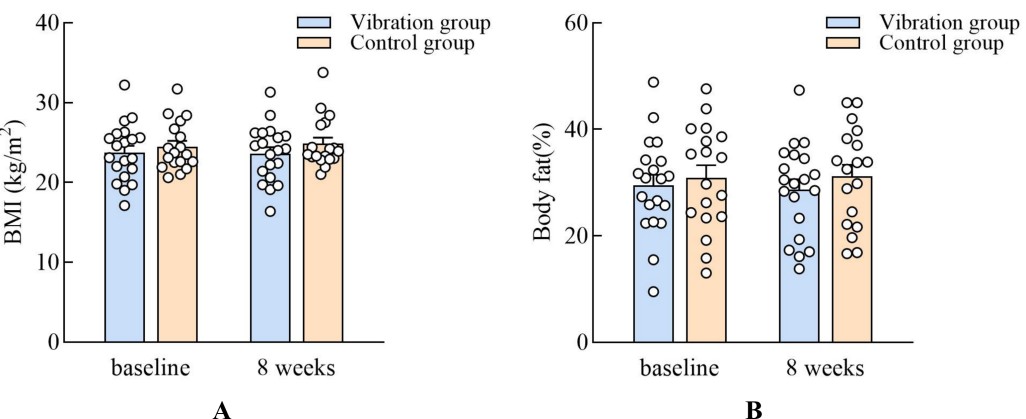

**Figure 6 Comparisons of the change after 8 weeks training in the BMI and body fat percentage.**

vibration frequency used was 8 ± 2 Hz (the present study used 5–13 Hz). A previous study showed that high-frequency vibration training had a greater impact than low-frequency vibration training on young people's muscles (*Sañudo et al., 2016*). Unlike the study by *Faes et al. (2018)* that focused on a population with a high degree of physical function, our study examined persons 75 years or older, a population with poorer physical function but with the potential for substantial improvement. According to the findings of the study by *Sañudo et al. (2016)*, vibration training enhances blood flow to the body, increases blood flow, quickens cellular metabolism, and boosts the body's capacity for sustained movement during exercise (*Sitjà-Rabert et al., 2015*).

The time needed to perform the TUG test after 8 weeks of low-frequency vibration was significantly reduced in older adults compared with both before the training and with the control group. These findings suggested that low-frequency vibration training improved the dynamic balance of older adults. *Tsuji et al. (2014)* reported that short-term whole-body vibration training improved functional mobility more than training without vibration stimulation in a TUG assessment; however, this benefit persisted only for 30 min (*Tsuji et al., 2014*).

Vibration training improves ankle and knee stability (*Moezy et al., 2008*), which can enhance postural control and is connected with greater neuromuscular control and decreased joint stiffness. The vibration produced by the vibration platform is transmitted through proprioceptive receptors, and this transmission of mechanical vibration triggers the tonic vibration reflex, a complex spinal and supraspinal neurophysiological response. This reflex in turn activates muscles and improves physical performance, and vibration-stimulated haptic feedback and intrinsic proprioception assist in controlling postural stability (*Fischer et al., 2019*). *Capicíková et al. (2006)* demonstrated that the muscles of the lower limbs of the leg can alter the state of postural balance during standing and are dependent on the activation and duration of proprioceptive stimulation. Vibration training stimulates the body's proprioceptors, promoting better balance by triggering brain plasticity and maintaining the body's postural position. This is so because during standing,

some muscles are engaged to a certain extent, but while vibrating, all muscles are always engaged.

We found that 8 weeks of low-frequency vibration training markedly increased the walking speed of older adults compared with both before the vibration training and with the control group. The improvement may depend on the individual's baseline level. For example, a previous review reported that vibration training increased walking speed among older adults by 36%, while walking speed in the untrained group increased by 18.1%, suggesting that the starting level of individuals may be a significant influencing factor (*Pollock, Martin & Newham, 2012*). Increases in single-leg support time following vibration training and the body's enhanced postural control are additional reasons why vibration training may improve walking. When the lower limbs are supported, walking becomes more similar to normal gait kinematics, and this stability forms the basis for the swing leg's forward motion during walking. This lower limb support encourages an increased walking pace (*Chan et al., 2012*). The ankle joint requires plantarflexion as the foot is forced upward off the vibration platform during walking. An improvement in lower extremity joint range of motion caused by vibration training has been demonstrated (*Szopa et al., 2021*). Therefore, improved joint range of motion and ankle muscle strength may also contribute to the increase in walking speed. Joint mobility, however, was not assessed in the study and should be investigated in future studies.

## Effects of low-frequency vibration training on the body composition

Vibration training did not show a statistically significant effect on BMI or body fat percentage. While a reduction in body fat mass was observed in the vibration group and a slight increase in the control group, these changes were not sufficient to conclude that vibration training effectively lowers BMI in older adults. Our findings are consistent with those of *Jo et al. (2021)* showing that vibration training had no appreciable impact on body fat percentage and BMI in older adults (*Jo et al., 2021*; *Reis-Silva et al., 2023*). However, most previous studies focused on older women; thus, gender should be investigated in future studies. *Fjeldstad et al. (2009)* (*Fjeldstad et al., 2009*)found that vibration training with a frequency of 15–40 Hz and an amplitude of three mm has a beneficial effect on body fat mass among postmenopausal women. Additionally, *Pérez-Gómez et al. (2020)* reported that vibration training (12–24 Hz, 3 times/week, three mm) decreased body fat percentage (4.4%) among middle-aged women. It is possible that vibration training alters the body's metabolic rate, resulting in an increase in energy expenditure and an inhibition of adipogenesis (*Cristi-Montero, Cuevas & Collado, 2013*) as well as changes plasma levels of staphylococcal or senescence marker protein 30 (SMP30), which is primarily involved in hepatic lipid regulation and lipid metabolism in women (*Kondo & Ishigami, 2016*). Additionally, several studies have shown a positive correlation between changes in the resistance index, mean velocity, and peak velocity of blood flow with changes in body composition after vibration training, demonstrating that changes in blood flow are linked to changes in fat mass (*Sañudo et al., 2013*). The present study showed that low-frequency vibration training had a significant impact on WHR and WC. Dietary factors also impact changes in body composition, but diet was not taken into account in this study.

### Study limitations

This study has limitations, including unmonitored physical activity and sedentary behavior, potential influence of control group activities, lack of participant blinding, absence of follow-up for long-term effects, and adherence rates of 80% in the vibration group and 72% in the control group, which may limit the interpretation and generalizability of the results. Finally, due to the high rate of data loss caused by health conditions or unforeseen events (*e.g.*, illness or death), intention-to-treat analysis was not applied.

## CONCLUSIONS

The findings of this randomized controlled trial indicated that 8 weeks of low-frequency vibration training significantly enhanced dynamic balance and maximum walking speed among older adults. The training may also have assisted in preserving their lower limb muscle strength, which can protect the capacity to sit and stand as the body ages. Vibration training had a positive impact on body composition, significantly reducing WC and WHR, which benefits overall physical health. Thus, long-term low-frequency vibration training may be a safe and effective tool for improving the health of older adults.

### Funding

This work was supported by a grant from The Program for Overseas High-level talents at Shanghai Institutions of Higher Learning (No. TP2020063). The funders had no role in study design, data collection and analysis, decision to publish, or preparation of the manuscript.

### Grant Disclosures

The following grant information was disclosed by the authors:
Shanghai Institutions of Higher Learning: TP2020063.

### Competing Interests

The authors declare there are no competing interests.

### Author Contributions

- Xiaohuan Tan conceived and designed the experiments, performed the experiments, analyzed the data, prepared figures and/or tables, authored or reviewed drafts of the article, and approved the final draft.
- Guiping Jiang conceived and designed the experiments, analyzed the data, authored or reviewed drafts of the article, and approved the final draft.
- Lei Zhang performed the experiments, prepared figures and/or tables, and approved the final draft.
- Dandan Wang performed the experiments, prepared figures and/or tables, and approved the final draft.
- Xueping Wu conceived and designed the experiments, authored or reviewed drafts of the article, and approved the final draft.

## Human Ethics

The following information was supplied relating to ethical approvals (i.e., approving body and any reference numbers):

The study was approved by the Ethics Committee of Shanghai University of Sport (102772022RT123).

## Data Availability

The raw measurements are available in the Supplemental File.

## Supplemental Information

Supplemental information for this article can be found online at http://dx.doi.org/10.7717/peerj.19263#supplemental-information.

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
