# Peer review of "Effects of low-frequency vibration training on walking ability and body composition among older adults: a randomized controlled trial"

_PeerJ, doi:10.7717/peerj.19263_

## Round 0.1 · original submission · Major Revisions

· Academic Editor

Major Revisions

I have now received evaluations from experts in the field. Although the reviewers were generally positive, they indicated significant concerns that require your attention before the manuscript can be accepted. Accordingly, I invite you to respond to the reviewers' comments and recommendations and to submit a revised version of your manuscript.

A final decision concerning the acceptance of your manuscript will be made after the original reviewers have examined the revisions and provided their assessment of the revised manuscript and your responses to the reviewers' critiques. Please be aware that this invitation does not guarantee acceptance of your manuscript.

Reviewer 1 ·

Basic reporting

English is clear, references are quite sufficient to explain the background behind the aim.
Article structure is well-reported. However, I suggest to revise the figures.

Experimental design

The experimental approach resemble the randomized control trial.

Validity of the findings

Long-term low-frequency vibration training may be a safe and effective tool for improving the health-related outcomes in older adults.

Additional comments

Abstract:
Line 33: The domain of the capacity to walk does not reflect the 30-s Chair Stand. Consider to reformulate the domain and report/discuss it as physical performance test.
Line 44: reformulate the sentence regarding BMI and body fat percentage.
Line 45: Conclusion does not exactly reflect the outcomes obtained. Please consider to reformulate it.

Introduction:
Line 95: references as a support to the hypothesis of the study should be implemented. Moreover, a section on the differences in whole-body vibration protocols already used, especially in this population should be pointed out (for instance: doi:10.14283/jarlife.2021.7; 10.1016/j.heliyon.2024.e35822; doi:10.1016/j.apmr.2023.04.002).

Materials and methods:
Line 101: Please consider to report "Study design" as a subheading. Moreover, how randomization was conducted should be reported and better highglight the blinding.
Line 126: If the aim of figure 2 is to synthesize the protocol, the caption as a support should be reformulated. Consider to add letters (e.g, a), b), c) as supporting reference for each section of the figure). Moreover, you may consider to merge it with the protocol reported in Table 1.
Lines 128-131: On which basis the intervention was selected? Please report a reference as a support of the protocol selected.
Line 137: The 30s CST test should not be reported in the walking ability test section. Therefore, this should be modified also in the discussion section.

Results:
Line 187: Table 3 is not readable. Please rearrange it.
In figure 6, if no statistical significance is reported, please consider to eliminate the p values. Please recheck also the other figures.

Reviewer 2 ·

Basic reporting

All my comments can be found in the attached PDF file

Experimental design

All my comments can be found in the attached PDF file

Validity of the findings

All my comments can be found in the attached PDF file

Additional comments

All my comments can be found in the attached PDF file

Annotated reviews are not available for download in order to protect the identity of reviewers who chose to remain anonymous.

---

## Round 0.2 · accepted · Accept

· Academic Editor

Accept

Please advise your co-authors of this decision as soon as possible. The referee reports are copied at the end of this email.

Reviewer 1 ·

Basic reporting

Authors addressed my comments and I think that now the article is suitable for publication.

Experimental design

no comment

Validity of the findings

no comment

Additional comments

no comment

Reviewer 2 ·

Basic reporting

The authors addressed all my suggestions. I think the paper is now ready for publication in PeerJ.

Experimental design

No comment

Validity of the findings

No comment

Additional comments

No comment